# Modeling and Assessment of Landslide Susceptibility of Dianchi Lake Watershed in Yunnan Plateau

**Guangshun Bai** [1,2,3] 🆔, **Xuemei Yang** [4], **Zhigang Kong** [1,2,3,*], **Jieyong Zhu** [1,2,3], **Shitao Zhang** [1,2,3,*] **and Bin Sun** [1,2,3] 🆔

1     Faculty of Land Resource Engineering, Kunming University of Science and Technology, Kunming 650093, China; baiguangshun@kust.edu.cn (G.B.); zhujieyong@kust.edu.cn (J.Z.); binsun@stu.kust.edu.cn (B.S.)
2     Key Laboratory of Geohazard Forecast and Geoecological Restoration in Plateau Mountainous Area, Ministry of Natural Resources of People's Republic of China (MNR), Kunming 650093, China
3     Key Laboratory of Geohazard Forecast and Geoecological Restoration in Plateau Mountainous Area in Yunnan Province, Kunming 650093, China
4     Yunnan Gaozheng Geo-Exploration Co., Ltd., Kunming 650041, China; yangxuemeilj@foxmail.com
*     Correspondence: zhigangkong@kust.edu.cn (Z.K.); taogezhang@hotmail.com (S.Z.)

**Abstract:** The nine plateau lake watersheds in Yunnan are important ecological security barriers in the southwest of China. The prevention and control of landslides are important considerations in the management of these watersheds. Taking the Dianchi Lake watershed as a typical research area, a comprehensive modeling and assessment process of landslide susceptibility was put forward. The comprehensive process was based on the weight of evidence (WoE) method, and many statistical techniques were integrated, such as cross-validation, multi-quantile cumulative Student's comprehensive weight statistics, independence testing, step-by-step modeling, ROC analysis, and ROC-based susceptibility zoning. In this paper, fourteen models with high accuracy and validity were established, and the AUC reached 0.83–0.87 and 0.85–0.88, respectively. In addition, according to the susceptibility zoning map compiled via the optimal model, 80% of landslides can be predicted in the very-high- and high-susceptibility areas, which only account for 19.58% of the study area. Finally, this paper puts forward strategies for geological disaster prevention and ecological restoration deployment.

**Keywords:** landslide susceptibility assessment; weight of evidence (WoE) method; modeling; Dianchi Lake watershed; Yunnan Plateau

## 1. Introduction

Yunnan Province is an important ecological security barrier in the southwest of China, and lakes are important ecological regions. The Yunnan Province government is promoting the ecological protection and restoration of nine plateau lake watersheds (Figure 1). The prevention of landslides is one of the important goals of these activities. In order to support this ongoing project, it is necessary to evaluate landslide susceptibility by taking the lake watershed as the unit, choosing the geological environment and human activity factors that may affect or control landslide susceptibility in the watershed, understanding the distribution of landslide susceptibility in the watershed, and guiding the formulation of targeted prevention and control countermeasures. Dianchi Lake is the largest of the nine plateau lakes in Yunnan Province, and Kunming City is in this watershed, where human engineering activities are relatively strong. The research on landslide susceptibility assessment (LSA) in the whole Dianchi Lake watershed is still relatively lacking. Generally speaking, it is reasonable to choose Dianchi Lake watershed as the study area.

Landslides in plateau mountainous areas will always be a problem because they affect people's lives, destroy the surface, and cause economic losses [1]. Identifying the dangerous areas related to landslides is an important part of disaster management [2],

and it is also an important foundation for promoting human safety, infrastructure development, and ecological environment protection [1]. LSA describes the spatial probability of landslides [3,4]. On the regional scale, the landslide susceptibility modeling method based on statistics is considered to be appropriate [2,5–7].

Statistical LSA is a supervised dichotomy problem, which can be solved via different classification methods [7]. About 163 different data-driven methods are applied to LSA [8], such as weight of evidence (WoE) [9,10], naïve Bayes (NB) [6,11], logistic regression (LR) [12–15], discriminant analysis (DA) [3,16], supported vector machines (SVM) [17,18], random forest (RF) [13,19–21], artificial neural networks (ANN) [22–24], and many others. These methods have their advantages and disadvantages. When dealing with sparse landslide datasets, a simple algorithm can usually provide better results [25]. In addition, the analysis should be kept as simple as possible so that we might obtain a deeper understanding of the effect when testing the new model [7].

In this study, we choose the WoE method, which is a moderately complicated data-driven method based on statistics [9,10]. It is a well-known and widely used statistical method that is used to estimate the relationship between observation data (landslide training inventory) and potential control factors (geological and geomorphological factors) [10,26]. It is widely used in landslide susceptibility mapping (LSM) [1,2,6,7,11,17,27–31] because it is easy to understand and has a strict mathematical foundation and theoretical system. Although WoE has been frequently used in LSA in recent decades [1,2,6,7,9,11,17,27–31], establishing how to optimize the modeling process to improve the accuracy and validation of the model is a problem worth exploring. Because WoE only uses discrete data, continuous raster data need to be classified [2,4,10]. However, there is no standardized factor data classification method. This is the other problem worth discussing. In addition, how to reduce the statistical errors caused by the randomness of landslides and related factors is also worth studying because landslides usually do not happen by accident [4,6,32].

This study focuses on LSA and LSM in Dianchi Lake watershed. It has outstanding application value, which aims to enhance our ability to assess the susceptibility of landslides and improve the corresponding consulting services for stakeholders involved in disaster reduction. Concerning the research content, on the one hand, the characteristics of landslide sensitive factors were clarified; on the other hand, the spatial distribution of landslide susceptibility was clarified, which provides important technical support for guiding ecological restoration and the deployment of landslide prevention and mitigation methods in plateau lake watersheds. Concerning the technical aspect, this paper puts forward a comprehensive process of LSA based on the WoE method, including (1) data preparation; (2) optimizing the compilation of datasets for factor classification based on a cumulative Student's comprehensive weight (sC) curve and WoE statistics; (3) screening modeling factors based on the cross-validation theory and AUC of single-factor analysis; (4) optimizing a high-performance model based on the step-by-step modeling; and (5) dividing landslide susceptibility areas based on ROC. This paper obtains the results of the LSM with excellent fitting and prediction performance (both AUC reached 0.87) based on the above process. The spatial distribution map of landslide susceptibility classification was compiled, and the strategies of geological disaster prevention were put forward.

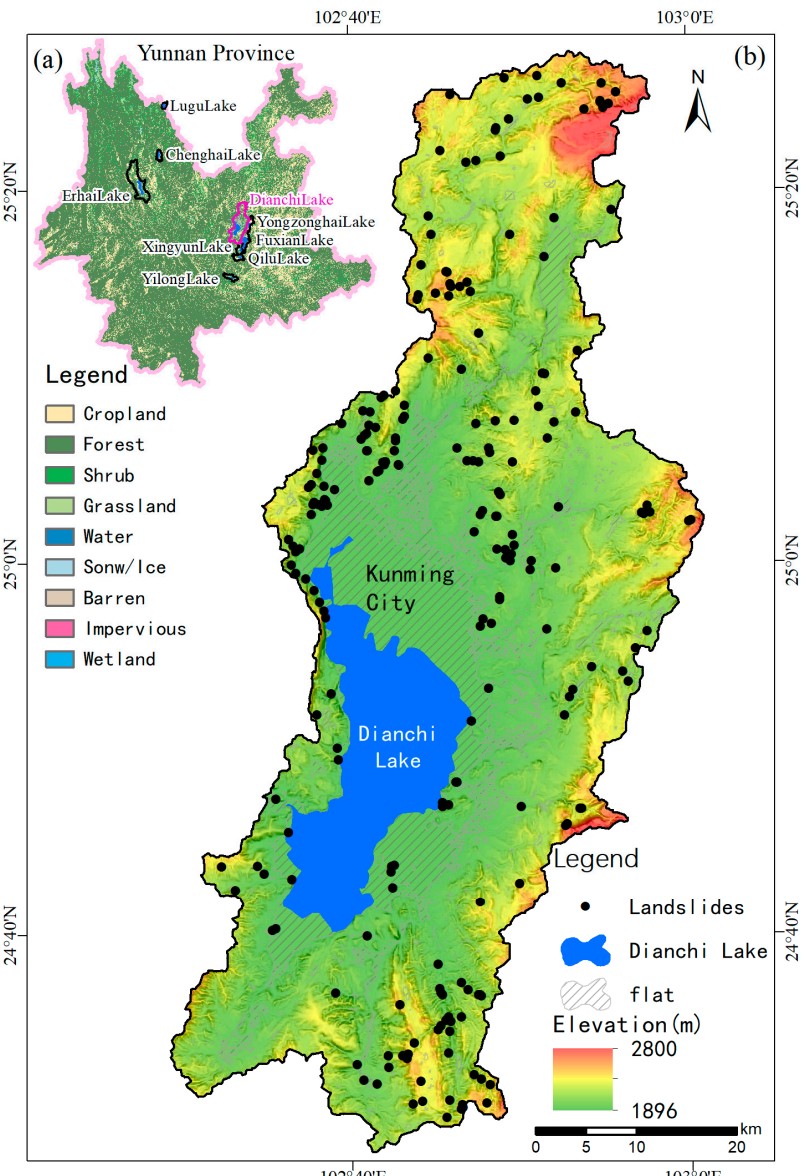

**Figure 1.** Study area. (**a**) The distribution of nine plateau lake watersheds in Yunnan, and the location of the Dianchi Lake watershed. The base map is the distribution map of land coverage types in Yunnan Province in 2020 [33]. (**b**) The distribution map of landslide points in Dianchi Lake watershed. The black points are landslides under investigation, the blue blocks are the water surface, and the gray diagonal lines are the areas with the attribute of "flat" [34,35]. The bottom picture is rendered via elevation and hill shade.

## 2. Study Area and Data

### 2.1. Study Area

The study area is the whole watershed of Dianchi Lake (Figure 1), covering an area of 2906.44 km². The climate belongs to the subtropical plateau monsoon climate, which is divided into a dry season and a rainy season. Most of the rainfall is between May and October, with an average annual rainfall of 1000 mm and an average temperature of 14.8 °C [36]. The water surface of Dianchi Lake and reservoir, the center of Kunming basin, the sub-basins, and some flat hilltops are not susceptible to landslides. Therefore, according to the result of the DEM classification [34,35], the actual analysis area in this paper is 2206.29 km² after deducting 700.15 km² of the "flat" category. This area is a lake basin and the mountainous terrain in the central part of Yunnan Plateau. The lake is

located in the southcentral region, rugged mountains are in the north, east, and south, and Xishan mountain, which has steep fault cliffs, is in the west. The elevation of this area ranges from 1896 m at the surface of Dianchi Lake to about 2800 m in the mountainous area, with a height difference of over 900 m. The steep mountainous terrain around the basin, the continuous and rapid river cutting, the heavy rainfall in the rainy season, and the man-made influence of downward slope cutting during road construction make this area prone to slope failure. According to the official regional geological survey report, the strata of different times were merged according to lithology. The loose gravel soil, sandstone, mudstone, shale, siltstone, basalt, limestone, and metamorphic rocks are distributed in the study area (Figure 2).

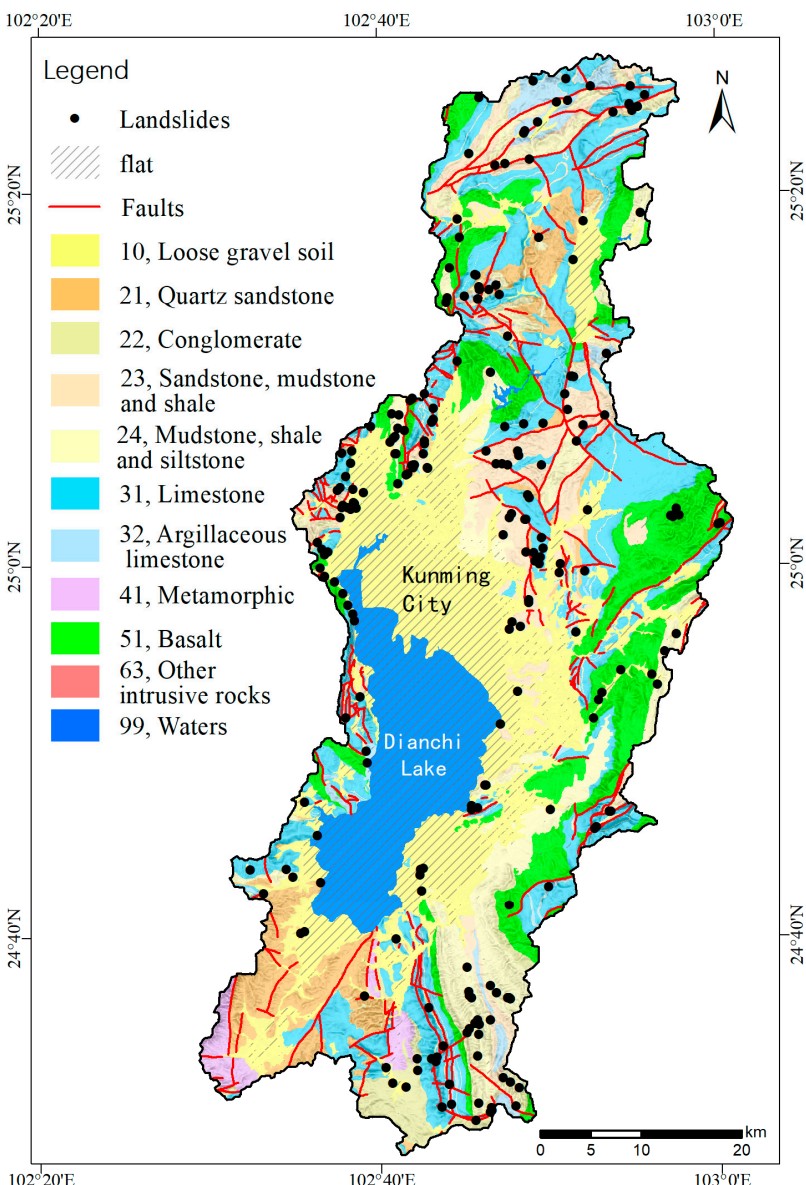

**Figure 2.** The geological map of lithology and faults.

## 2.2. Landslides and Data Preparation Based on Random Sampling

The study area has a good working history in landslide surveys, and it is the key monitoring and prevention area of landslides in Yunnan Province. Through field investigation, the historical landslides list was checked and revised, and a total of 228 landslides were included in the landslides list analyzed in this paper (Figure 1).

We adopted the cross-validation technique to prepare the data. The cross-validation technique is a basic technique employed to evaluate the uncertainty of statistical data and models using test datasets that do not involve model training [4,37,38]. Figure 3 briefly summarizes the compilation process of the landslide dataset. (1) We divided all landslide data (ALL) into a training dataset (TRN) containing 158 landslides and a test dataset (TST) containing 70 landslides using random sampling tools. TRN and TST are not duplicated, which account for about 70% and 30% of ALL, respectively. TRN is used to calibrate the model, and TST is used to evaluate the performance of the model. (2) To estimate the model variables that depend on the sample size, we used the random sampling tool to generate 100 random sub-samples with TST size from TRN, and some landslides in different random sub-samples were allowed to be repeated, forming a training data subset trn.

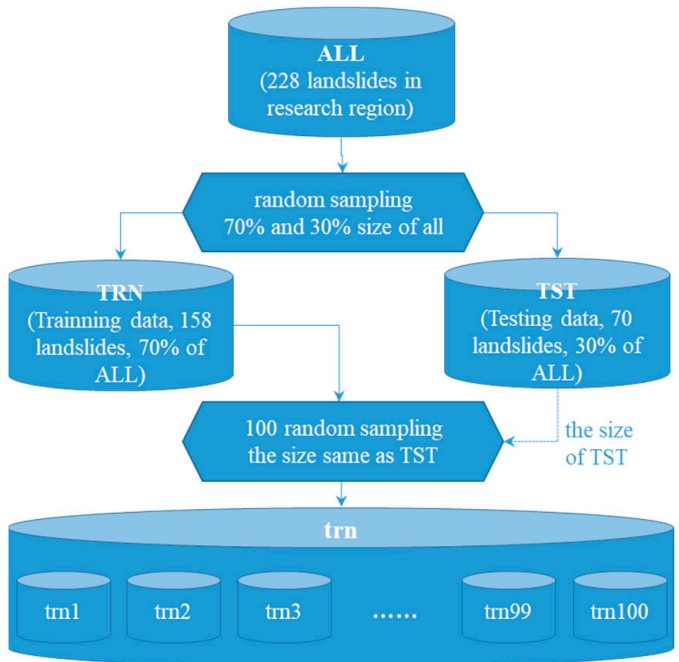

**Figure 3.** Process flowchart of cross-validation landslide dataset compilation based on random sampling.

### 2.3. Factor Data

According to the characteristics of the study area, the availability data, and previous research, landslide control factors can be roughly divided into different groups [6,11]. Table 1 lists the landslide control factors which have been compiled. At first, we did not rule out available or main factors that are easy to deduce, as these factors may help explain landslide susceptibility. These factors have been used in many other studies, and descriptions of these factors can be found in a large number of studies [2,11,19,39–42]; so, this article did not elaborate upon this further. In Table 1, we briefly explained the significance of these factors in LSA.

**Table 1.** Sources and significances of the factors.

| No. | General Category | Factors | Significance | Source and Compilation Method |
|---|---|---|---|---|
| 1 | Geologic | Distance to faults (dF) | Destruction of the stability of the rock mass structure | The fault structural lines came from the 1:200,000 geological map of Kunming; using QGIS to compile Euclidean distance grid |
| 2 | | Lithology (Lth) | Lithological types of slope rock and soil | 1:200,000 geological map of Kunming |

**Table 1.** *Cont.*

| No. | General Category | Factors | Significance | Source and Compilation Method |
|---|---|---|---|---|
| 3 | | CLCD | The 30 m annual land cover dataset in China | The 30 m annual land cover dataset and its dynamics in China 2019 (CLCD) [33] |
| 4 | Land cover | Land cover (LC) | The 10 m land cover | ESA WorldCover 10 m 2020 v100 [43] |
| 5 | | Normalized difference vegetation index (NDVIlog) | | China 30 m Annual NDVI Maximum Dataset (2021) [44] as the log value |
| 6 | Anthropogenic | Distance to roads (dRD) | Road cutting or vehicle vibration | Data come from OSM (OpenStreetMap, 2021); using QGIS to compile the Euclidean distance grid |
| 7 | Morphometric terrain parameters | Elevation (Elv) | Climate, vegetation, and potential energy | NASADEM [45], the resolution of which is ~30 m |
| 8 | | Aspect (Asp) | Solar insolation, flora and fauna distribution and abundance [1] | Compilation using SAGA GIS via DEM [45] |
| 9 | | Plan curvature (CPlan) | Converging, diverging flow, soil water content, and soil characteristics [1] | Compilation using SAGA GIS via DEM [45], with value $\times 10^6$ |
| 10 | | Profile curvature (CProf) | Flow acceleration, erosion/deposition, and geomorphology [1] | Compilation using SAGA GIS via DEM [45], with value $\times 10^6$ |
| 11 | | Tangential curvature (CTang) | Erosion/deposition [1] | Compilation using SAGA GIS via DEM [45], with value $\times 10^6$ |
| 12 | Morphometric terrain parameters | Topographic Position Index (TPI) | Quantifies topographic heterogeneity and erosion [46] | Compilation using SAGA GIS via DEM [45] |
| 13 | | Terrain Ruggedness Index (TRI) | Quantifies topographic heterogeneity and erosion [47] | Compilation using LSAT PM [4] via DEM [45] |
| 14 | | Roughness (Rou) | Quantifies topographic heterogeneity and erosion | Compilation using LSAT PM [4] via DEM [45] |
| 15 | | Relative slope position (RSP) | | Compilation using LSAT PM [4] via DEM [45] |
| 16 | | Slope (SL) | Stress field is related to slope | Compilation using SAGA GIS via DEM [45] |
| 17 | | Flow path length (FPL) | River erosion | Compilation using SAGA GIS via DEM [45] |
| 18 | | Flow Accumulation (FAlog) | Runoff velocity, runoff volume, and potential energy | Compilation using SAGA GIS via DEM [45] as the log value |
| 19 | | Height above nearest drainage (HAND) | River erosion, runoff velocity, runoff volume, and potential energy [48,49] | Compilation using SAGA GIS via DEM [45] |
| 20 | | Horizontal HAND (HANDH) | River erosion, runoff velocity, runoff volume, and potential energy [48,49] | Compilation using SAGA GIS via DEM [45] |
| 21 | Water-related | Vertical HAND (HANDV) | River erosion, runoff velocity, runoff volume, and potential energy [48,49] | Compilation using SAGA GIS via DEM [45] |
| 22 | | Distance to channel network (dCN) | River erosion. | Compilation using SAGA GIS via DEM [45] |
| 23 | | Stream power index (SPIlog) | River erosion [50] | Compilation using SAGA GIS via DEM [45] as the log value |
| 24 | | Topographic wetness index (TWI) | Moisture content of soil [50–52] | Compilation using SAGA GIS via DEM [45] |
| 25 | | SAGA Wetness Index (TWISAGA) | Moisture content of soil [52,53] | Compilation using SAGA GIS via DEM [45] |

## 3. Methods

### 3.1. Weights-of-Evidence Method (WoE)

The weights of a single factor are superimposed on the linear model to obtain the complete landslide sensitivity model [1,10,26,28]. WoE was first introduced in the late 1980s, and it was used for the application of geological science based on GIS, mainly for the mapping of mineral potential [10,26,54–56]. $D$ is defined as the unit with landslides, $\overline{D}$ as the unit without landslides, $B$ as the unit in the evidence factor area, $\overline{B}$ as the unit outside the evidence factor area, $P(\mid)$ as the conditional probability symbol, and $N(\cdots)$ as the grid pixels number. WoE considers two kinds of weights and posterior probability [2,6,10,11,26,55,56]:

$$W^+ = ln\frac{P(B|D)}{P(B|\overline{D})} = ln\left(\frac{N(B \cap D)}{N(B \cap D) + N(\overline{B} \cap D)} \middle/ \frac{N(B \cap \overline{D})}{N(B \cap \overline{D}) + N(\overline{B} \cap \overline{D})}\right), \quad (1)$$

$$W^- = ln\frac{P(\overline{B}|D)}{P(\overline{B}|\overline{D})} = ln\left(\frac{N(\overline{B} \cap D)}{N(B \cap D) + N(\overline{B} \cap D)} \middle/ \frac{N(\overline{B} \cap \overline{D})}{N(B \cap \overline{D}) + N(\overline{B} \cap \overline{D})}\right). \quad (2)$$

The weight symbols $W^+$ and $W^-$ do not represent the mathematical meaning of numerical values; rather, they represent the presence (positive) and absence (negative) of feature classes in a given raster cell. According to the above formula, a positive logical value indicates the positive impact of a given variable, a negative logical value indicates the negative impact, and a logical value of zero represents no influence.

The posterior probability is an index of susceptibility, the higher numerical value means higher susceptibility, and a lower numerical value means lower susceptibility. The formula for calculating the posterior probability is as follows: $P = O/1 + O = exp(F)/(1 + exp(F))$, $F = \sum_{i=0}^{n} W_i^{K(i)} + lnO(D)$, $O(D) = N(D)/N(\overline{D})$, where $K(i)$ is "+" when the $i$-th evidence factor layer exists, and $K(i)$ is "−" when it does not exist; $W_i$ is the weight of the existence or non-existence of the $i$-th evidential factor.

In order to evaluate the spatial correlation strength between single factors, landslide, and the performance of the model, the receiver operating characteristic curve (ROC) algorithm is used in this paper, which is a technique employed to visualize and evaluate the performance of the classifier by describing the ratio of the true positive rate (sensitivity) to the false positive rate (1-specificity) [57]. The area under the ROC curve (AUC) provides a quantitative index by which to compare the advantages and disadvantages. Generally speaking, the AUC is excellent when it is greater than 0.8, the AUC is good when it is 0.7–0.8, the AUC is moderate when it is 0.6–0.7, and the AUC is common when it is smaller than 0.6.

### 3.2. Main Analysis Process

In this paper, a comprehensive evaluation process of landslide susceptibility based on WoE is proposed, which mainly includes (Figure 4): (1) data preparation (see Sections 2.2 and 2.3 for details); (2) optimizing the compilation of datasets for factor classification (see Section 3.4 for details); (3) screening modeling factors; (4) step-by-step modeling to optimize high-performance model; and (5) dividing landslide susceptibility level zones based on the ROC of model.

### 3.3. WoE Statistical Process

Our WoE statistical process integrates cross-validation technology and traditional WoE statistical technology (Figure 5). It can solve the statistical error caused via the randomness of landslides and factors [4,6,32], and it has a deeper understanding than the traditional WoE statistics which only use all the landslide data. In this paper, trn (containing 100 subsets) was used for statistics. For each factor, the statistical process was repeated 100 times. We calculate the mean weight of each factor category (WoE_trn)

and its corresponding statistical values, such as variance and standard deviation. ROC is used to evaluate the classification ability of each factor for each statistical data point graphically. There are two advantages to this statistical process [6]: first, based on its estimated variance, it can better represent the general uncertainty of the sensitivity model; second, for classified data, it can determine whether the significance weight has accidental characteristics, or whether it can be reproduced from different random samples, which is more likely to be causality.

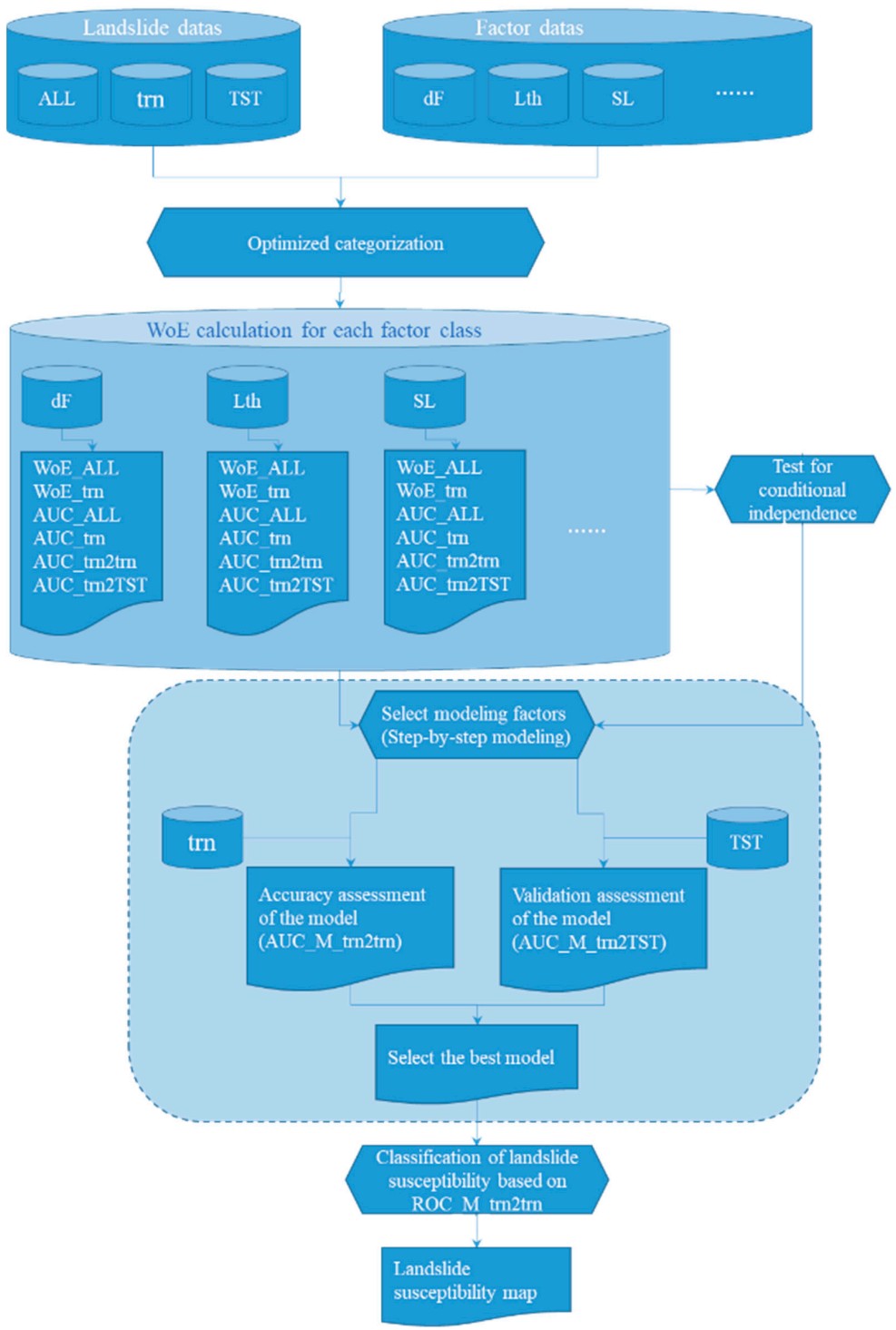

**Figure 4.** Flowchart of the improved WoE landslide susceptibility assessment.

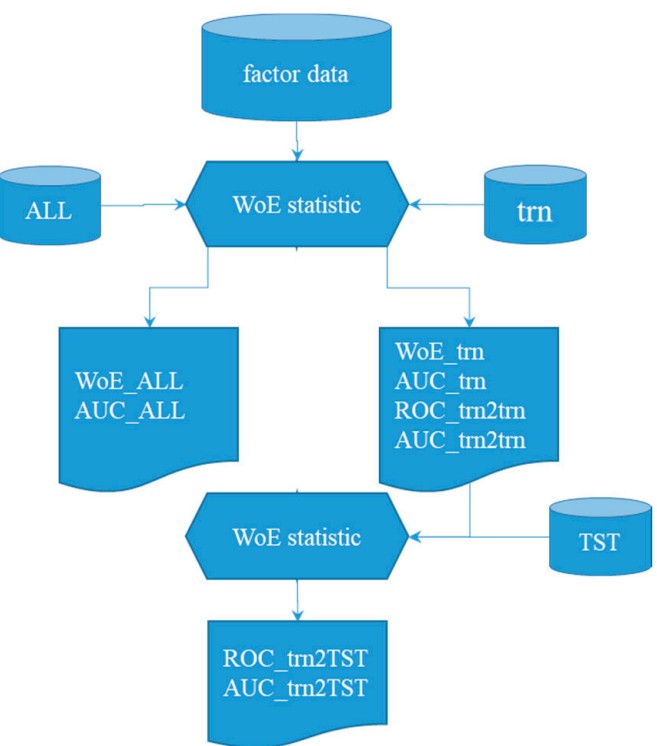

**Figure 5.** Process flowchart of single-factor WoE statistic. WoE_ALL, sC_ALL, and AUC_ALL are the weight, sC, and AUC calculated on ALL, respectively; WoE_trn, sC_trn, and AUC_trn are the mean weight, sC, and AUC calculated 100 times on trn, respectively; ROC_trn2trn and AUC_trn2trn are the single-factor accuracy assessment indexes modeled by single-factor weight WoE_trn and fit to trn; and ROC_trn2TST and AUC_trn2TST are the single-factor validity assessment indexes modeled by single-factor weight WoE_trn and fit to TST.

The trn is used to evaluate the accuracy performance of the model, and the TST is used to evaluate the validation performance of the new data prediction model [7,31] (Figure 6). If the ROC curve based on the TST falls within the range of the ROC curve based on the trn (representing MSE), this shows that the accuracy and validation of the model can be good; otherwise, the model may be over-fitted.

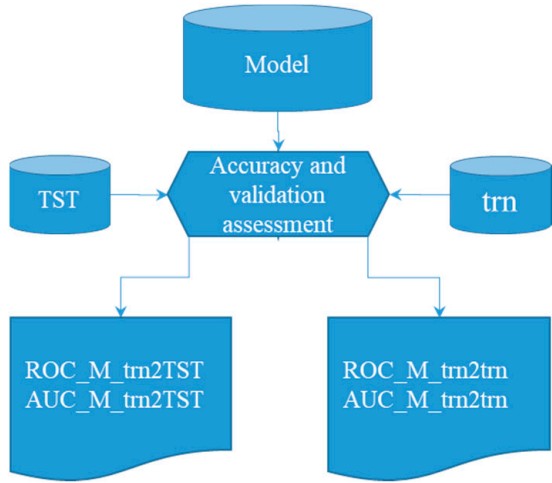

**Figure 6.** Process flowchart of accuracy and validation assessment of models.

### 3.4. Optimization Process of Single-Factor Categorization

Because WoE uses discrete data, it is necessary to classify continuous single-factor data discretely, which will lead to a discontinuity of factor weights. The determination of the traditional single-factor discrete classification number and classification threshold is subjective. This paper puts forward a single-factor classification optimization process (Figure 7), and the main steps are as follows.

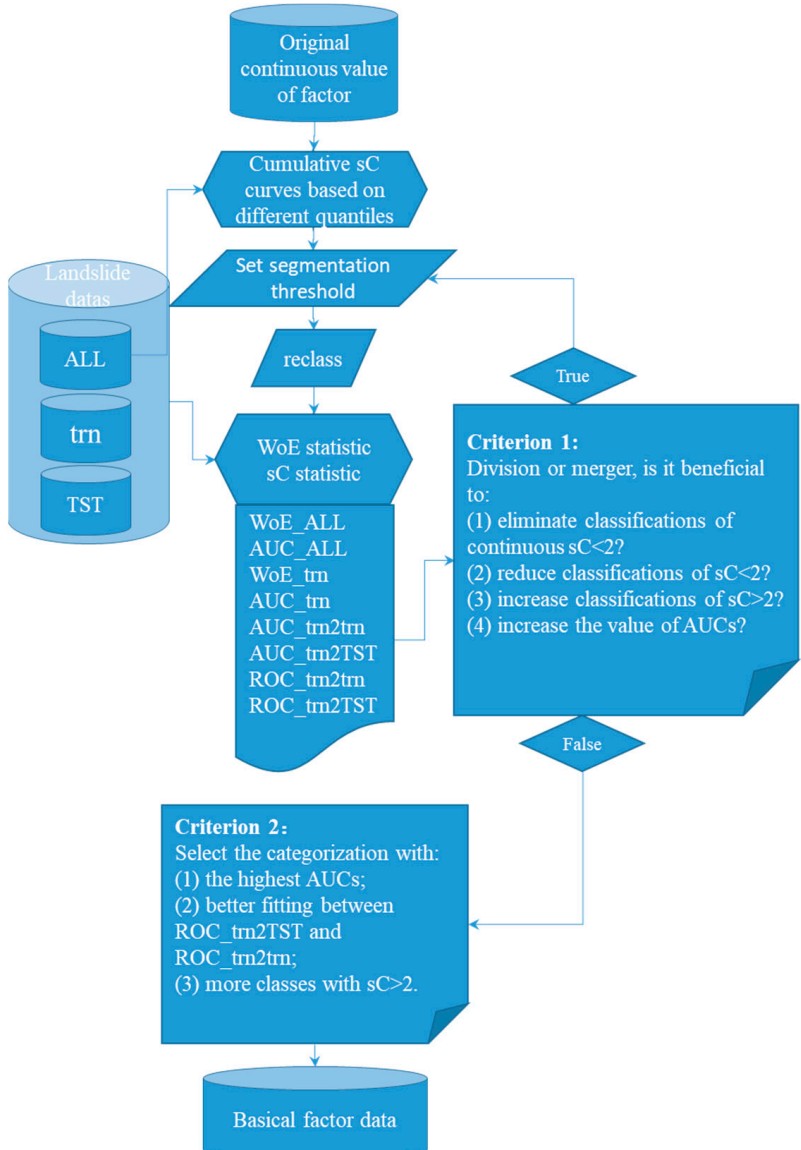

**Figure 7.** Process flowchart of factor classification optimization strategy based on the cumulative *sC* curve and WoE statistics.

First, a cumulative *sC* curve is generated. This method involves subdividing the continuous numerical single-factor raster into multiple classes according to the quantile and calculating the weight and corresponding variance for each class. The difference between the two weights—that is, the comprehensive weight—and the correlation between the quantitative factor and landslides are calculated as follows [26]: $C = W^+ - W^-$. A confidence measure defined via contrast divided by its standard deviation is introduced, which is similar to the Student's comprehensive weight (*sC*). The *sC* is relatively large when the standard deviation is small, so the results are more reliable. When the test values of *sC* are 1.96 and 2.326, confidence levels are 97.5% and 99%, respec-

tively [10,26,55]. $sC = C/\sigma_C = C/\sqrt{\sigma_{W^+}^2 + \sigma_{W^-}^2}$, $\sigma_{W^+}^2 = 1/N(B \cap D) + 1/N(B \cap \overline{D})$, $\sigma_{W^-}^2 = 1/N(\overline{B} \cap D) + 1/N(\overline{B} \cap \overline{D})$, where $\sigma_C$, $\sigma_{W^+}$, and $\sigma_{W^-}$ are standard deviations of $C$, $W^+$, and $W^-$, respectively. A new discrete distance category is defined using the accumulated $sC$ [6]. As long as the weight value is positive, $sC$ should be increased; when the weight is close to zero, it should be flat; when the weight is negative, it should be decreased. Therefore, the shape of the cumulative $sC$ curve shows its maximum value at the position where it is expected to have the greatest influence. If there is more than one maximum value, this indicates the distortion effect of another variable [6].

Based on the cumulative $sC$ curve, the classification and segmentation thresholds are set, the factors are reclassified, and the reclassified factor data are subjected to single-factor WoE statistics (Section 3.3).

Then, set a new trial segmentation threshold is obtained and we repeat the above steps.

Finally, we suggest determining the best classification according to two criteria (Criterion 1 and Criterion 2). Criterion 1: the division or merger beneficial to (1) eliminating classifications of continuous $sC < 2$; (2) reducing classifications of $sC < 2$; (3) increasing classifications of $sC > 2$; or (4) increasing the value of AUCs. After several rounds of trial calculation, the optimal classification is determined according to Criterion 2: select the best categorization with (1) the highest AUCs; (2) the better fitting between ROC_trn2TST and ROC_trn2trn; and (3) the more classes with $sC > 2$.

## 4. Results

### 4.1. Cumulative sC Statistical Curve of Continuous Single Factor

ALL and six quantiles, namely, 100, 80, 60, 40, 20, and 10, were used to calculate the continuous numerical factors via the sub-process in Section 3.4. The statistical curve of cumulative sC (Figure 8) reveals the correlation between continuous numerical factors and the spatial distribution of landslides in different quantiles in detail, which not only reflect the changing trend of cumulative $sC$ but also show the details of cumulative $sC$ change.

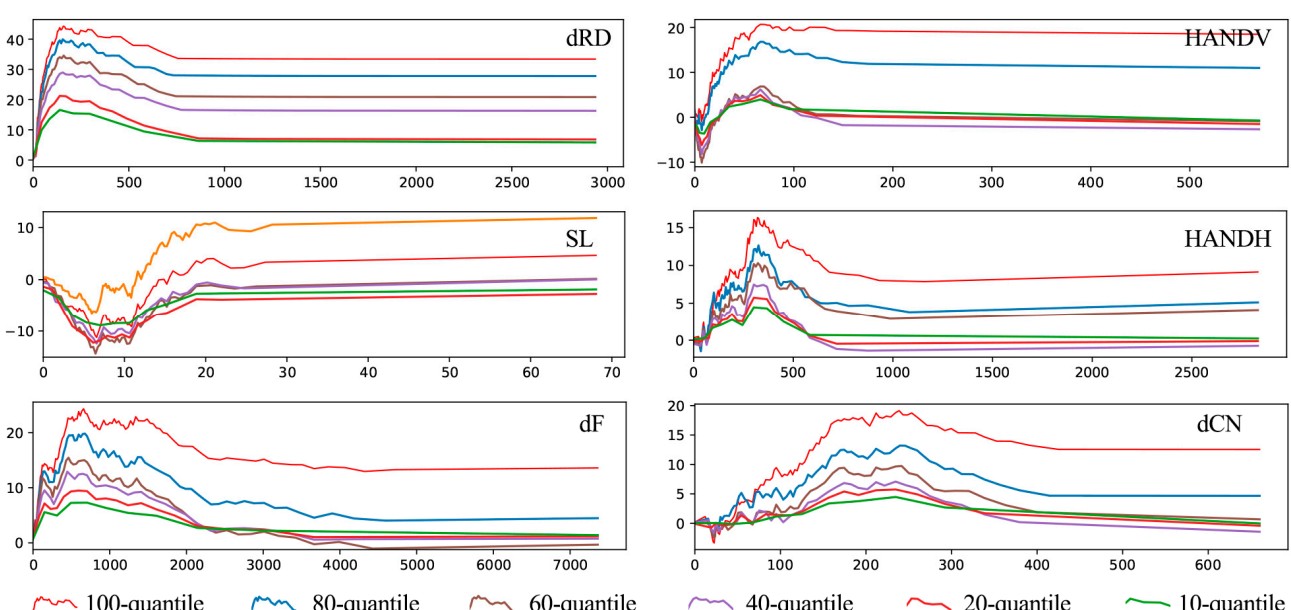

**Figure 8.** The cumulative $sC$ statistical curve of six factors according to six quantiles of 100, 80, 60, 40, 20, and 10.

In Figure 8, the curves of dRD, SL, and dF on the left are simpler than those of HANDV, HANDH, and dCN on the right, and the secondary fluctuation is smaller, indicating that dRD, SL, and dF have a strong spatial correlation with landslides and are less affected by other factors. The segmentation value can be further extracted from the cumulative

trend of weight reflected by the slope and continuity of the curve. Specifically, for the dRD, the positive weight at 157.42 m becomes negative; the 0–157.42 m curve has a large rising slope, which indicates that this is the key segment of landslide susceptibility. For the SL, the 10.83–21.10° segment is the key slope gradient that can easily induce landslides. For the dF, the positions of 121 m and 460 m are the key positions with positive and negative weight changes; 0–121 m and 262–460 m are the rising sections of the curve, and the secondary fluctuation is small, which indicates that this is the key area in which to induce landslide. For the HANDV, the weight changes positively and negatively at 6.93 m and 66.60 m, respectively, which are the key dividing points. The section ranging between 6.93–66.60 m is the key section that affects the landslide susceptibility. There are many secondary fluctuations on the HANDH curve, indicating that the spatial correlation between this factor and landslide is affected by other factors. For the dCN, there are also many secondary fluctuations in the curve, but generally speaking, 22.33–174.57 m is a rising section with a large slope, which has a strong correlation with the occurrence of landslides.

*4.2. Results of Single-Factor WoE Analysis*

Landslides usually do not happen by accident; they are unevenly distributed in different factors and factor categorizations [2,32]. After implementing the technical processes in Sections 3.3 and 3.4, this paper obtained the evidence weight and sensitivity strength analysis results of each factor (Figures 9–18).

(1)    Geological Factors

The dF is divided into five categories. Figure 9 shows that many landslides have been found in class 1 and class 3, and the positive weight is very high. As the distance increases, it becomes less and less easy to slide. The third diagram shows the error distribution caused by spatial random effects and also reveals the stability of positive weights for class 1 and class 3. This result reflects that faults make joints and cracks in the nearby rock mass develop, which is conductive to the occurrence of landslides. The position 460 m away from the fault is a key demarcation point, and an area less than 460 m is especially conducive to landslides. The spatial correlation between dF and landslides is moderate, with the mean and the range of the AUC_trn being 0.63 and 0.58–0.68 (the fourth picture).

Rock strata is divided into five categories. Most landslides occurred in the class 24 (mudstone, shale, and siltstone) and class 23 (sandstone, mudstone, and siltstone) categories. The statistical results of cross-validation technology (the third figure) helped us further confirm the distribution of positive weight. The spatial correlation between Ltd. and landslide is moderate, with the mean and the range of the AUC_trn being 0.63 and 0.58–0.70 (the fourth picture).

(2)    Land Cover Factors

Landslides do not easily occur in forest areas because the roots of trees reinforce the slopes. The statistical results of NDVIlog and CLCD factors obtained the same understanding. Among these two factors, the areas with low NDVIlog value (<3.81) and grassland are the most prone to landslides. The spatial correlation between NDVIlog and landslide is high, with the mean and the range of the AUC_trn being 0.66 and 0.61–0.71 (the fourth picture).

(3)    Anthropogenic Factors

Slope cutting in road construction and vehicle vibration lead to landslides. According to the cumulative comparative weight analysis, dRD was classified into 9 classes (Figure 13). The first and third pictures show that the area of <157.42 m is prone to landslides. The spatial correlation between dRD and landslide is very high, with the mean and the range of the AUC_trn being 0.71 and 0.68–0.75 (the fourth picture).

(4)    Morpho-metric Terrain Parameters

The statistical results show that the spatial correlations between SL (Figure 14), RSP (Figure 15), TRI (Figure 16), and Rou (Figure 17) and landslides are moderately high, while the spatial correlation between CProf (Figure 18) and landslides is moderately low. The results of SL show that the weights of class 5 (10.83–11.65°), class 10 (25.60–28.27°), and class 11 (28.27–39.98°) are very high, and landslides are prone to occur in these areas.

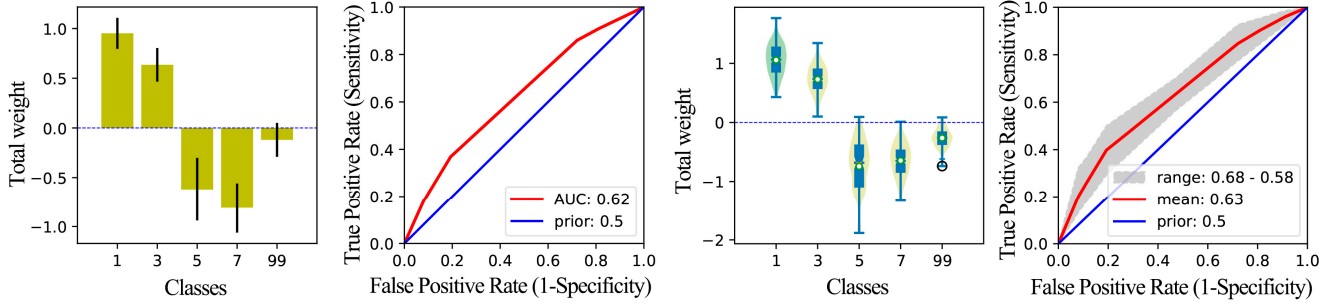

**Figure 9.** Graphical result of WoE for the factor dF. Class 1 is 0–121 m; class 3 is 262–460 m; class 5 is 657–864 m; class 7 is 1355–2317 m; and class 99 is other ranges. The first picture is the C histogram of factor classification based on statistics of ALL, and the black vertical line is the error bar of C. The second picture presents the ROC_ALL and AUC_ALL based on statistics of ALL. The third picture is a C violin-box diagram of factor classification based on statistics of trn with 100 subsets. The fourth picture presents the ROC and AUC, which have been counted 100 times based on trn with 100 subsets, where the red line is the mean ROC and the gray band is the ROC range.

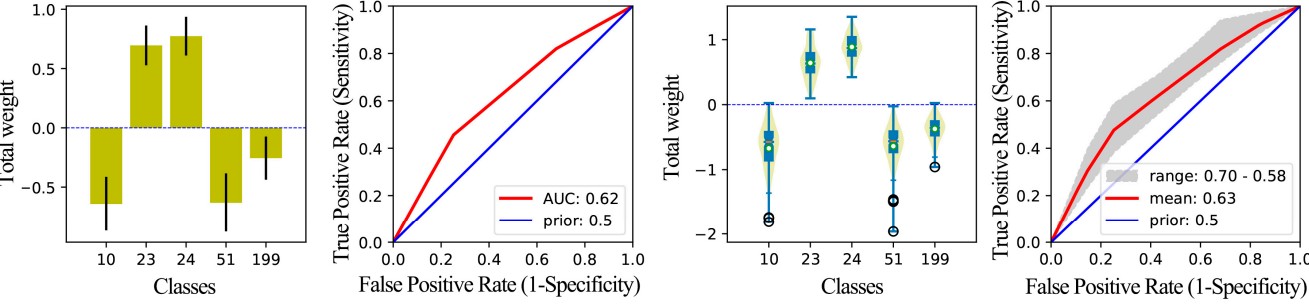

**Figure 10.** Graphical result of WoE for the factor Lth. Class 10 is loose gravel soil; class 23 is sandstone, mudstone, and shale; class 24 is mudstone, shale, and siltstone; class 51 is basalt; and class 199 is other lithologic strata, including limestone and metamorphic rocks.

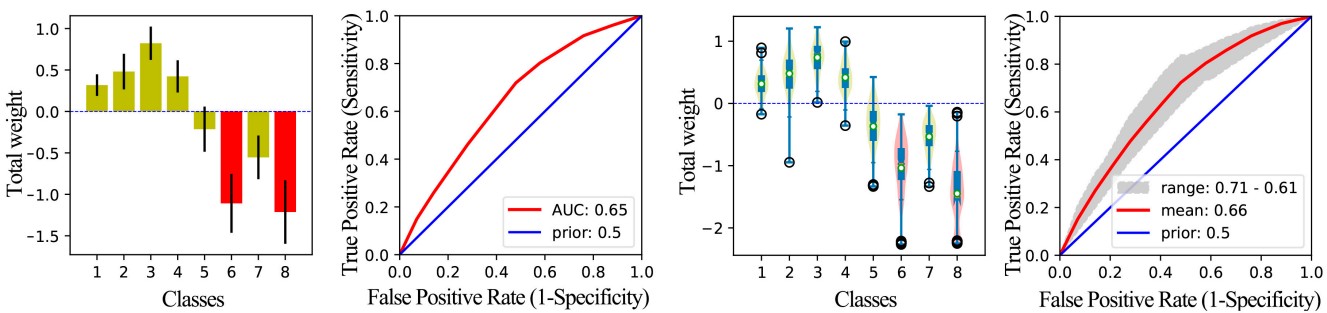

**Figure 11.** Graphical result of WoE for the factor NDVIlog. Class 1 is 2.79–3.64; class 2 is 3.64–3.71; class 3 is 3.71–3.76; class 4 is 3.76–3.81; class 5 is 3.81–3.84; class 6 is 3.84–3.85; class 7 is 3.85–3.88; and class 8 is 3.88–3.99.

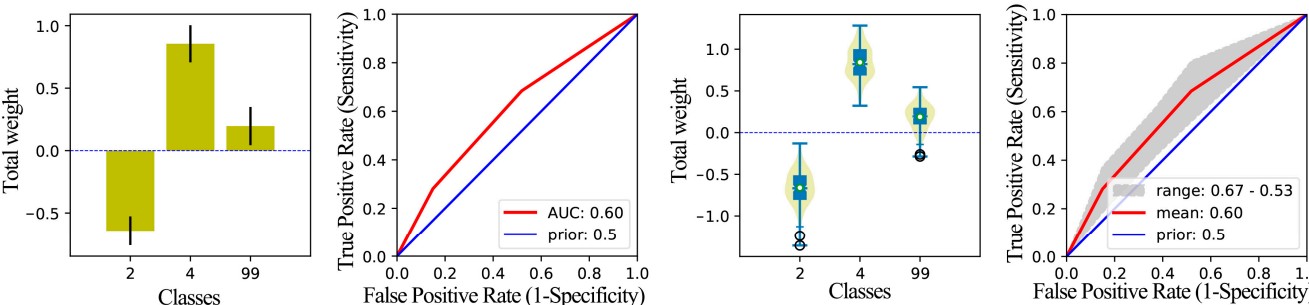

**Figure 12.** Graphical result of WoE for the factor CLCD. Class 2 is forest; class 4 is grassland; and class 99 is others (cropland, shrub, barren, impervious, wetland).

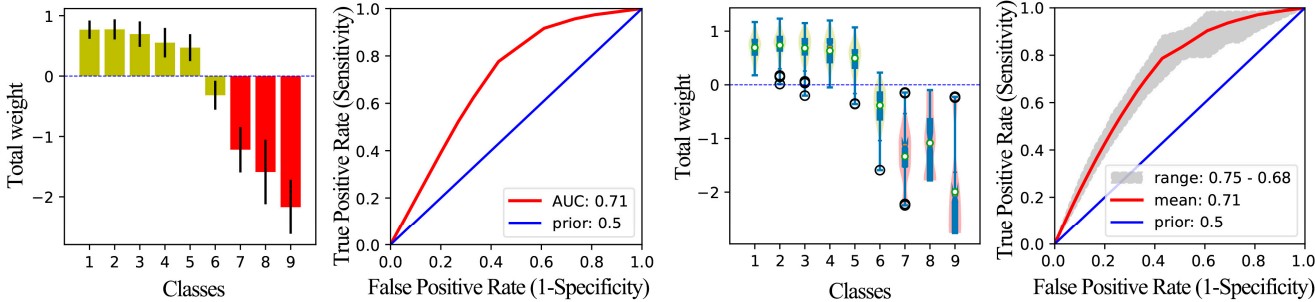

**Figure 13.** Graphical result of WoE for the factor dRD. Class 1 is 0–22.81 m; class 2 is 22.81–44.56 m; class 3 is 44.56–71.39 m; class 4 is 71.39–99.68 m; class 5 is 99.68–157.42 m; class 6 is 157.42–306.85 m; class 7 is 306.85–458.95 m; class 8 is 458.95–602.39 m; and class 9 is 602.39–2936.07 m.

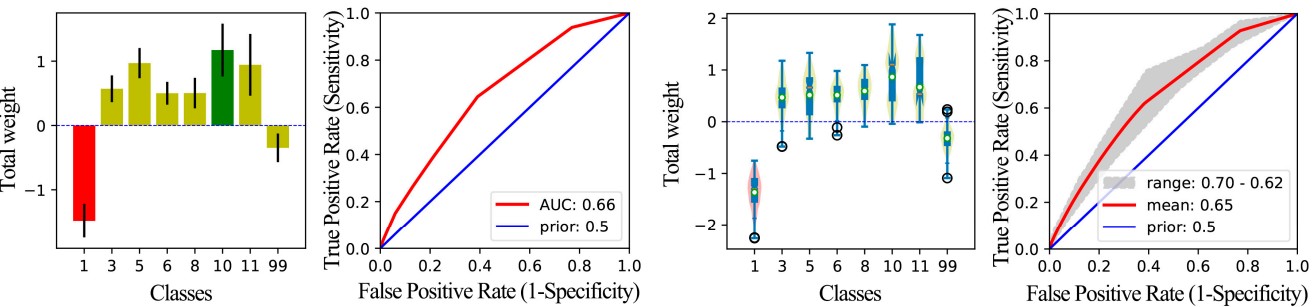

**Figure 14.** Graphical result of WoE for the factor SL. Class 1 is 0–4.12°; class 3 is 6.44–7.65°; class 5 is 10.83–11.65°; class 6 is 11.65–16.13°; class 8 is 17.12–21.10°; class10 is 25.60–28.27°; class 11 is 28.27–39.98°; and class 99 is other slopes.

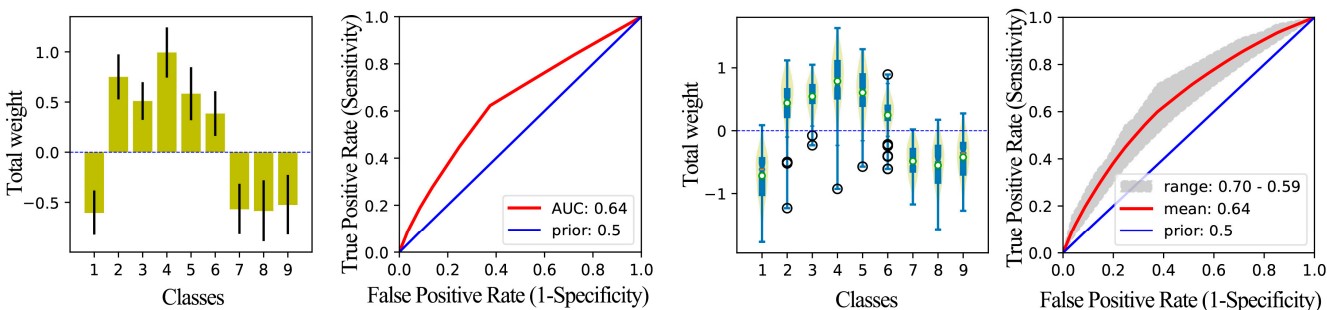

**Figure 15.** Graphical result of WoE for the factor RSP. Class 1 is 0–0.01; class 2 is 0.01–0.02; class 3 is 0.02–0.05; class 4 is 0.05–0.06; class 5 is 0.06–0.08; class 6 is 0.08–0.14; class 7 is 0.14–0.29; class 8 is 0.29–0.45; and class 9 is 0.45–1.02.

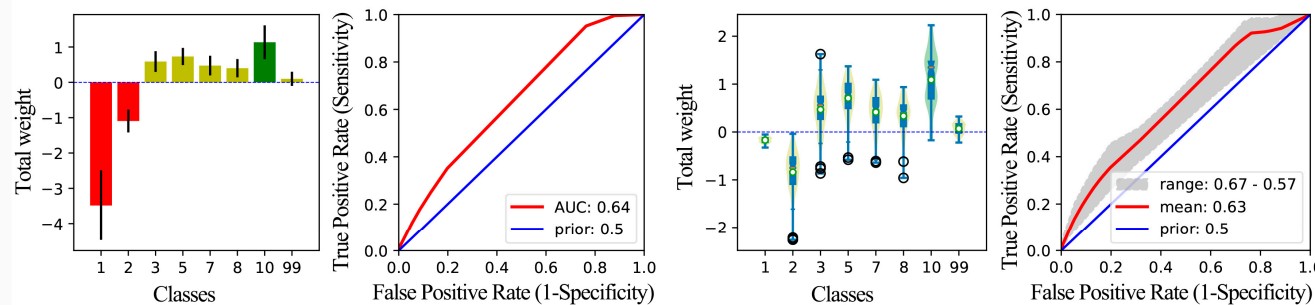

**Figure 16.** Graphical result of WoE for the factor TRI. Class 1 is 0.00–11.58 m; class 2 is 11.58–20.62 m; class 3 is 20.62–22.98 m; class 5 is 41.98–45.47 m; class 7 is 48.89–52.50 m; class 8 is 52.50–58.39 m; class 10 is 112.52–125.38 m; and class 99 is others in the range of 0–447.60 m.

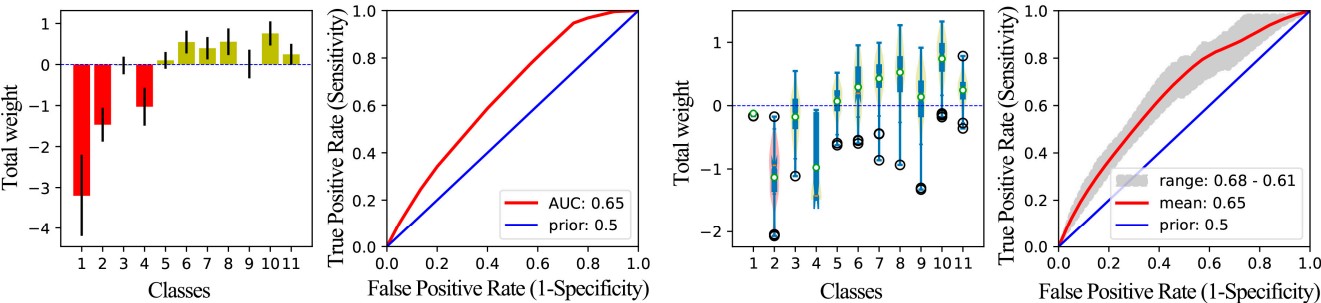

**Figure 17.** Graphical result of WoE for the factor Rou. Class 1 is 0.00–8.93; class 2 is 8.93–16.53; class 3 is 16.53–24.95; class 4 is 24.95–28.88; class 5 is 28.88–40.73; class 6 is 40.73–44.33; class 7 is 44.33–49.50; class 8 is 49.50–52.52; class 9 is 52.52–57.22; class 10 is 57.22–62.32; and class 11 is 62.32–398.73.

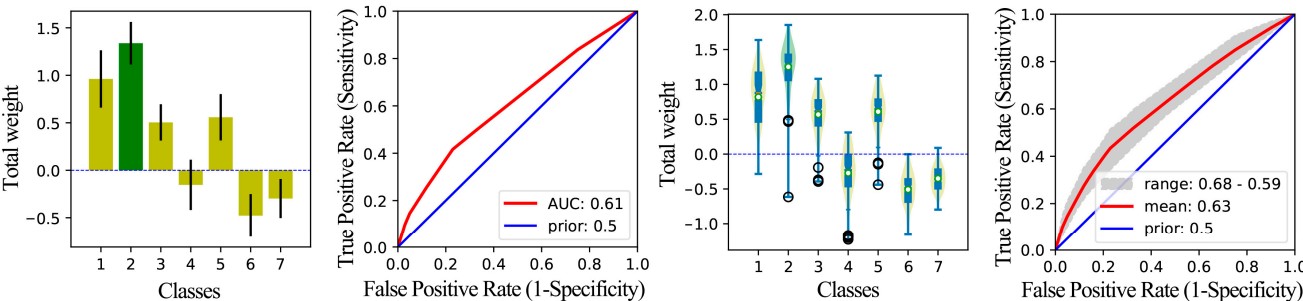

**Figure 18.** Graphical result of WoE for the factor Cprof. Class 1 is $-12{,}611.46 \sim -4084.50$ ($\times 10^{-6}$); class 2 is $-4084.50 \sim -2981.60$ ($\times 10^{-6}$); class 3 is $-2981.60 \sim -1533.30$ ($\times 10^{-6}$); class 4 is $-1533.30 \sim -973.62$ ($\times 10^{-6}$); class 5 is $-973.62 \sim -686.55$ ($\times 10^{-6}$); class 6 is $-686.55 \sim 37.07$ ($\times 10^{-6}$); and class 7 is $37.07 \sim 10596.92$ ($\times 10^{-6}$).

(5)    Water-related Factors

The spatial correlation between HANDV and landslide is moderately high (Figure 19), while the spatial correlation between HANDH (Figure 20) and dCN (Figure 21) and landslide is moderate.

The results (Figures 9–21) show 13 factors with AUC ≥ 0.6 (Figure 22). AUC reflects the strength of spatial correlation between factors and landslides. From the AUC of each factor in the Figure 22, we can see the difference of spatial correlation strength. The order of these factors from high to low is as follows: dRD, HANDV, NDVIlog, SL, RSP, TRI, Rou, Lth, dF, HANDH, Cprof, dCN, and CLCD.

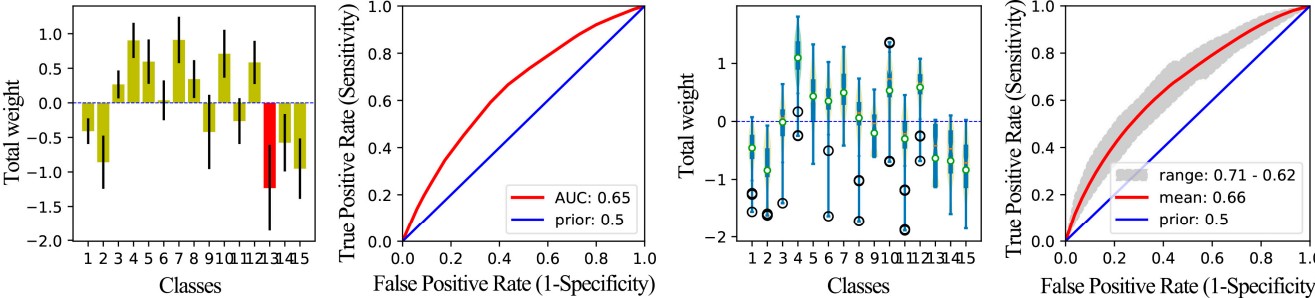

**Figure 19.** Graphical result of WoE for the factor HANDV. Class 1~class 15 are divided by 0 m, 4.15 m, 6.93 m, 13.03 m, 15.61 m, 17.89 m, 24.11 m, 26.22 m, 34.53 m, 37.77 m, 41.57 m, 55.48 m, 66.60 m, 77.37 m, 101.59 m, and 570.01 m.

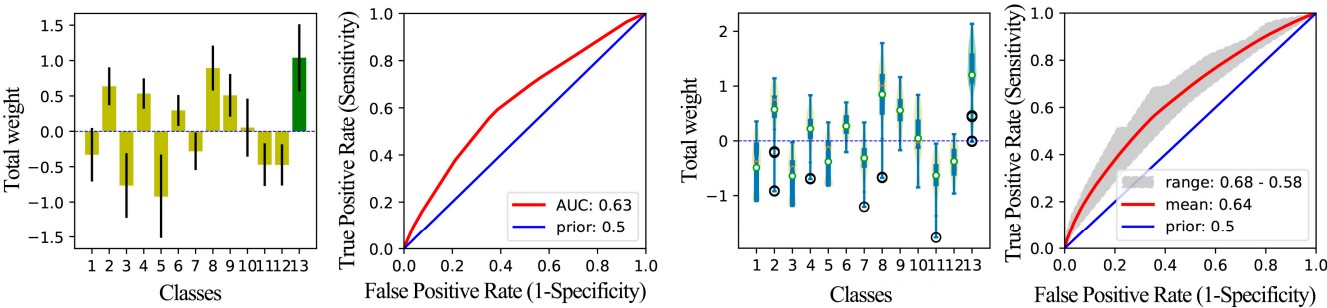

**Figure 20.** Graphical result of WoE for the factor HANDH. Class 1~class 13 are divided by 0 m, 38.06 m, 49.60 m, 65.22 m, 100.45 m, 115.44 m, 184.98 m, 1255.91 m, 271.86 m, 302.28 m, 323.25 m, 439.08 m, 1176.82 m, and 2831.14 m.

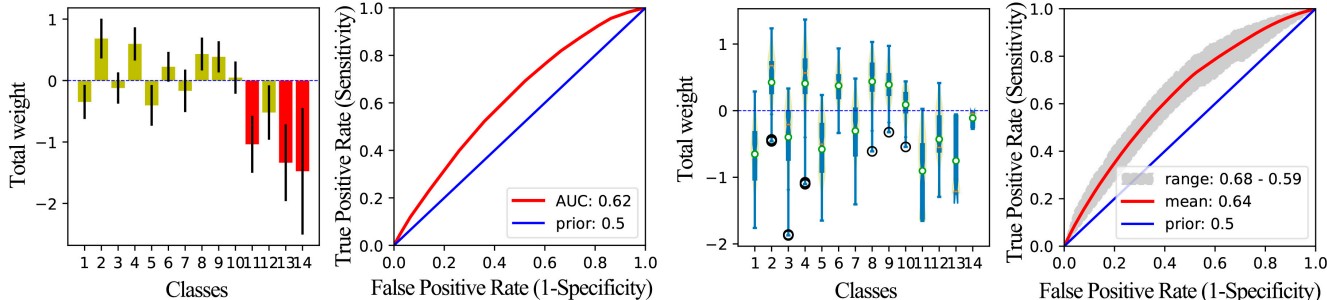

**Figure 21.** Graphical result of WoE for the factor dCN. Class 1~class 14 are divided by 0 m, 22.33 m, 24.98 m, 40.21 m, 49.85 m, 67.45 m, 94.96 m, 113.16 m, 134.62 m, 174.57 m, 240.09 m, 279.41 m, 320.53 m, 394.72 m, and more than 394.72 m.

We also analyzed the categorizations of factors with $W^+ \geq 0.6$ (Figure 23), which may be the key part of landslide susceptibility. Figure 23 shows the comparison of the spatial correlation between different factor categorizations and landslides.

### 4.3. Test Results for Conditional Independence

Strong correlation datasets may lead to incorrect estimations of factor contribution and expansion of the estimated probability value [58]. Chi-square-based contingency analysis is performed on the classified data based on the raster [4,11], according to Pearson's C and Cramer's V, to measure the correlation between discrete datasets.

| Factor | AUC_trn2trn | AUC_trn2TST | AUC_trn |
|--------|-------------|-------------|---------|
| dRD | 0.71 | 0.70 | 0.71 |
| HANDV | 0.65 | 0.65 | 0.66 |
| NDVIlog | 0.65 | 0.64 | 0.66 |
| SL | 0.64 | 0.68 | 0.65 |
| RSP | 0.63 | 0.66 | 0.64 |
| TRI | 0.63 | 0.64 | 0.63 |
| Rou | 0.63 | 0.62 | 0.65 |
| HANDH | 0.62 | 0.61 | 0.64 |
| Lth | 0.63 | 0.60 | 0.63 |
| dF | 0.63 | 0.60 | 0.63 |
| dCN | 0.62 | 0.60 | 0.64 |
| CProf | 0.62 | 0.61 | 0.63 |
| CLCD | 0.60 | 0.61 | 0.60 |

**Figure 22.** Thirteen factors with AUCs $\geq$ 0.6 and their AUC values.

| Factor | classID | class | W+_ALL | W+_trn | sC_trn |
|--------|---------|-------|--------|--------|--------|
| dRD | 1 | 0–22.81m | 0.671 | 0.610 | 430.9 |
| dRD | 2 | 22.81–44.56m | 0.673 | 0.643 | 29.2 |
| dRD | 3 | 44.56–71.39m | 0.639 | 0.629 | 51.0 |
| HANDV | 4 | 13.03–15.61m | 0.847 | 1.020 | 70.7 |
| HANDV | 7 | 24.11–26.22m | 0.876 | 0.480 | 9.5 |
| HANDV | 10 | 37.77–41.57m | 0.679 | 0.509 | 72.2 |
| NDVIlog | 3 | 3.71–3.76 | 0.751 | 0.681 | 45.3 |
| SL | 5 | 10.83–11.65° | 0.873 | 0.458 | 31.0 |
| SL | 10 | 25.60–28.27° | 1.115 | 0.826 | 32.2 |
| SL | 11 | 28.27–39.98° | 0.893 | 0.636 | 6.6 |
| RSP | 4 | 0.05–0.06 | 0.926 | 0.726 | 14.7 |
| TRI | 5 | 41.98–45.47m | 0.723 | 0.677 | 58.2 |
| TRI | 10 | 112.52–125.38m | 1.164 | 1.122 | 32.4 |
| Rou | 10 | 57.22–62.32m | 0.717 | 0.701 | 24.8 |
| HANDH | 8 | 255.91–271.86m | 0.847 | 0.801 | 16.5 |
| HANDH | 13 | 1176.82–2831.14m | 1.016 | 1.173 | 31.9 |
| Lth | 24 | Mudstone, shale and siltstone | 0.604 | 0.680 | 61.5 |
| dF | 1 | 0–121.25m | 0.808 | 0.870 | 62.7 |
| dCN | 2 | 22.33–24.98m | 0.672 | 0.414 | 5.2 |
| CProf | 1 | $-12611.46 - -4084.5$ ($\times 10^{-6}$ m$^{-1}$) | 0.873 | 0.744 | 196.1 |
| CProf | 2 | $-4084.50 - -2981.60$ ($\times 10^{-6}$ m$^{-1}$) | 1.207 | 1.120 | 78.9 |
| CLCD | 4 | Grassland | 0.633 | 0.625 | 193.8 |

**Figure 23.** Factor classification with $W^+ \geq 0.6$.

Figure 24 combines the statistical results of two correlation indexes, Pearson's C and Cramer's V, which are located in the upper right half and the lower left half of the heat map, respectively. The results show that according to Pearson's C index, Rou, and TRI (0.81) and Rou and SL (0.71) are strongly related factor pairs. However, according to Cramer's V, the correlation among the factors involved in statistics is not strong (<0.60). The correlation between dF, HANDH, and dCN and all other factors is very low. Elevation and its derived TRI, Rou, RSP, and SL have a slight relationship.

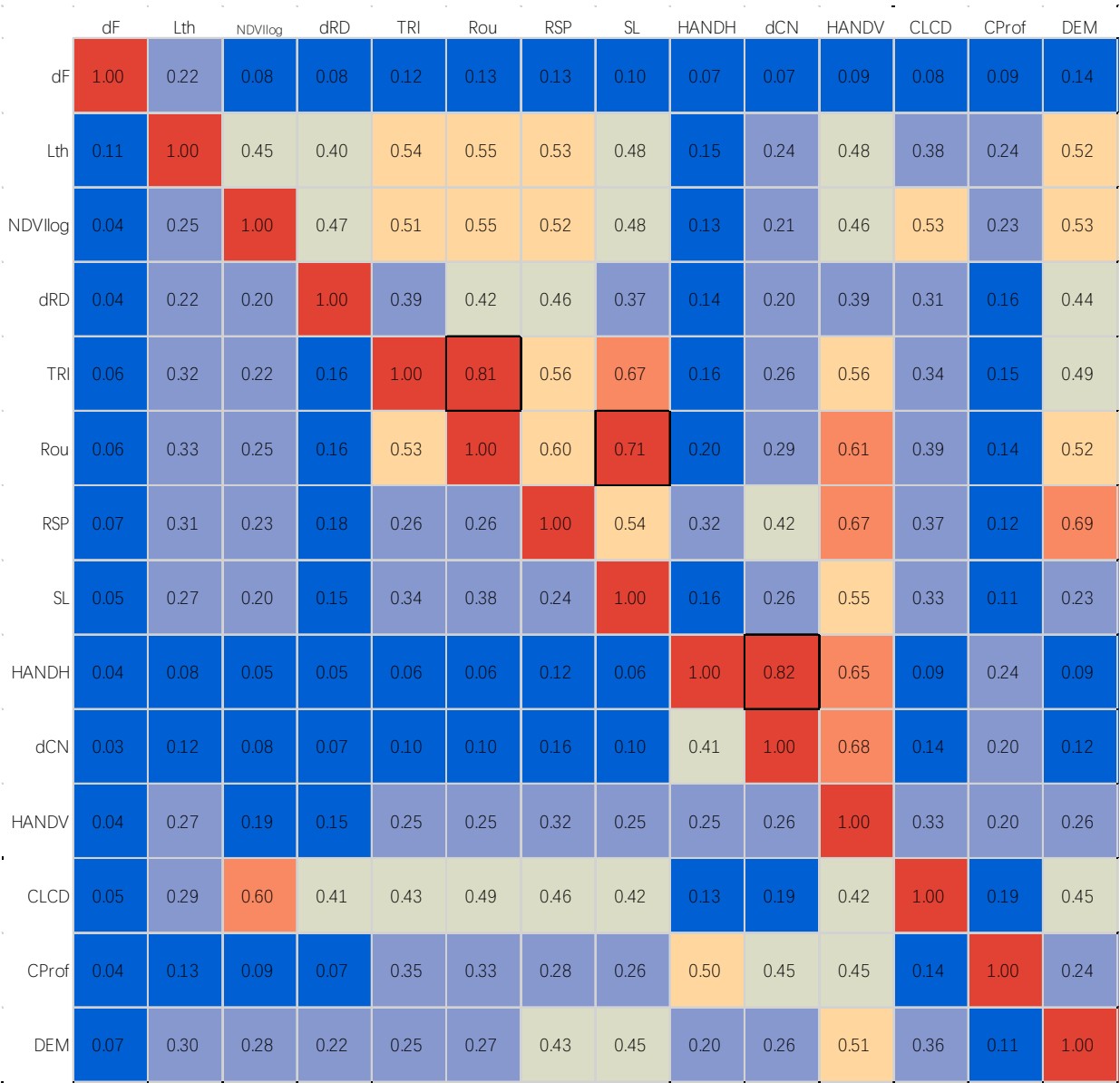

**Figure 24.** Test results for conditional dependence. The upper right half represents the Pearson's C results, and the factors with a strong correlation indicated by >0.7 are designated by black circles, such as Rou and TRI (0.81), Rou and SL (0.71), and dCN and HANDH (0.82). The lower left presents the Cramer's V results.

### 4.4. Step-by-Step Modeling Results of Landslide Susceptibility

According to AUCs and conditional dependencies, factors are sorted and combined. The model M6 is based on the combination of factors with high AUCs. Then, we try to add follow-up factors into the new model in turn, and evaluate the fitting performance, uncertainty, and the prediction performance of the new model via ROC_M and AUC_M. We discard factors that cannot improve the AUC_M or improve the consistency of ROC_M.

As shown in Figures 25 and 26, the success rate of the model M11 is represented by the ROC calculated via trn, and its AUC is ~0.87. The model M11 is the best model. The AUC of the prediction rate calculated via TST was ~0.87 too. Both of them are high, being within the range of excellent classification models. The results also show that the M11 has excellent fitting and prediction performance and has not been over-fitted.

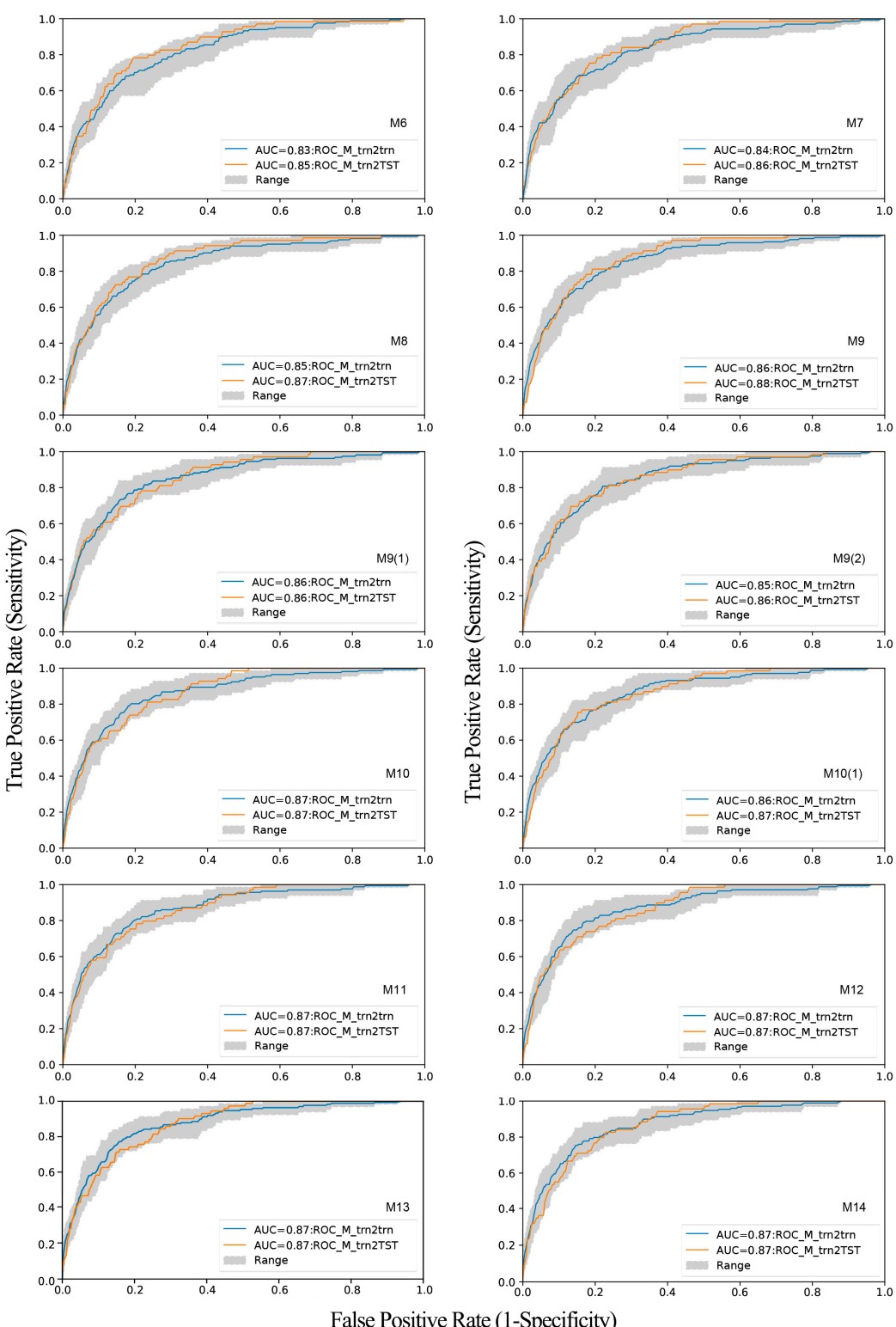

**Figure 25.** Accuracy and validity assessment of the models. Accuracy assessment of the models of susceptibility to landslides with the ROC_trn2trn of models (the blue line and the grey range). The total weights for the models were based on trn, and the performance of the models was evaluated using trn. One hundred iterations were carried out. The blue line is the mean ROC_M of 100 iterations. The grey range marks the model uncertainty based on the ROCs' MSE for 100 iterations. Test of validity of the models with the ROC_M_trn2TST (the orange line). The total weight maps were based on trn, and the validation was assessed using TST.

| Models | AUC_trn2TST | AUC_trn2trn | ROC fitting score | ROC fitting assessment | Fctors |
|---|---|---|---|---|---|
| M14 | 0.87 | 0.87 | 10 | Good | dF+Lth+CLCD+NDVIlog+dRD+CProf+TRI+ Rou+RSP+DEM_+SL+HANDH+dCN+HANDV |
| M13 | 0.87 | 0.87 | 9 | Slightly larger in the middle and right | dF+Lth+NDVIlog+dRD+CProf+TRI+Rou+ RSP+SL+HANDH+dCN+HANDV |
| M12 | 0.87 | 0.87 | 9 | Slightly larger on the left and right | dF+Lth+NDVIlog+dRD+CProf+TRI+Rou+ RSP+SL+HANDH+dCN+HANDV |
| M11 | 0.87 | 0.87 | 10 | Good | dF+Lth+NDVIlog+dRD+TRI+Rou+RSP+SL+ HANDH+dCN+HANDV |
| M10 | 0.87 | 0.87 | 9 | Slightly larger in the middle and right | dF+Lth+NDVIlog+dRD+TRI+Rou+RSP+SL+ HANDH+HANDV |
| M10(1) | 0.87 | 0.86 | 10 | Good | dF+NDVIlog+dRD+TRI+Rou+RSP+SL+HANDH+ dCN+HANDV |
| M9 | 0.88 | 0.86 | 9 | Slightly larger on the left | dF+NDVIlog+dRD+TRI+Rou+RSP+SL+HANDH+ HANDV |
| M9(1) | 0.86 | 0.86 | 10 | Good | Lth+NDVIlog+dRD+TRI+Rou+RSP+SL+HANDH+ HANDV |
| M9(2) | 0.86 | 0.85 | 10 | Good | NDVIlog+dRD+TRI+Rou+RSP+SL+ HANDH+dCN+HANDV |
| M8 | 0.87 | 0.85 | 10 | Good | NDVIlog+dRD+TRI+Rou+RSP+SL+ HANDH+HANDV |
| M7 | 0.86 | 0.84 | 10 | Good | NDVIlog+dRD+TRI+Rou+RSP+SL+ HANDV |
| M6 | 0.85 | 0.83 | 10 | Good | NDVIlog+dRD+TRI+RSP+SL+HANDV |

**Figure 26.** Comparison of validity and accuracy (AUCs) of models.

### 4.5. Landslide Susceptibility Mapping Results

Based on the ROC_M_trn2trn of model M11, we have compiled the landslide susceptibility zoning map (Figure 27). This method uses the success rate to determine that the cumulative landslide area exceeds the cumulative area that is considered vulnerable [59], which can improve the readability of the map. Very-high-susceptibility areas (VHS) comprise only 5.05% of the study area and contain 50% of the landslides. High-susceptibility areas (HS) comprise 14.53% of the study area and contain 30% of the landslides (Figure 27, Table 2). Medium-susceptibility areas (MS), low-susceptibility areas (LS), and very-low-susceptibility areas (VLS) comprise 28.23%, 32.55%, and 19.64% of the study area, respectively, and contain 15%, 4%, and 1% of landslides. Therefore, HS and VHS contain 80% of the landslides and only comprise 19.58% of the study areas. These characteristics of the landslide susceptibility zoning map represent the potential of M11 for the first-order prediction of landslides in this landscape.

**Table 2.** Statistical table of landslide susceptibility zoning area.

| Sub-Regions | Area of Sub-Regions (%) | Total Area of Sub-Regions (%) | Landslides (%) | Total Landslides (%) |
|---|---|---|---|---|
| VHS | 5.05 | 5.05 | 50 | 50 |
| HS | 14.53 | 19.58 | 30 | 80 |
| MS | 28.23 | 47.81 | 15 | 95 |
| LS | 32.55 | 80.36 | 4 | 99 |
| VLS | 19.64 | 100 | 1 | 100 |

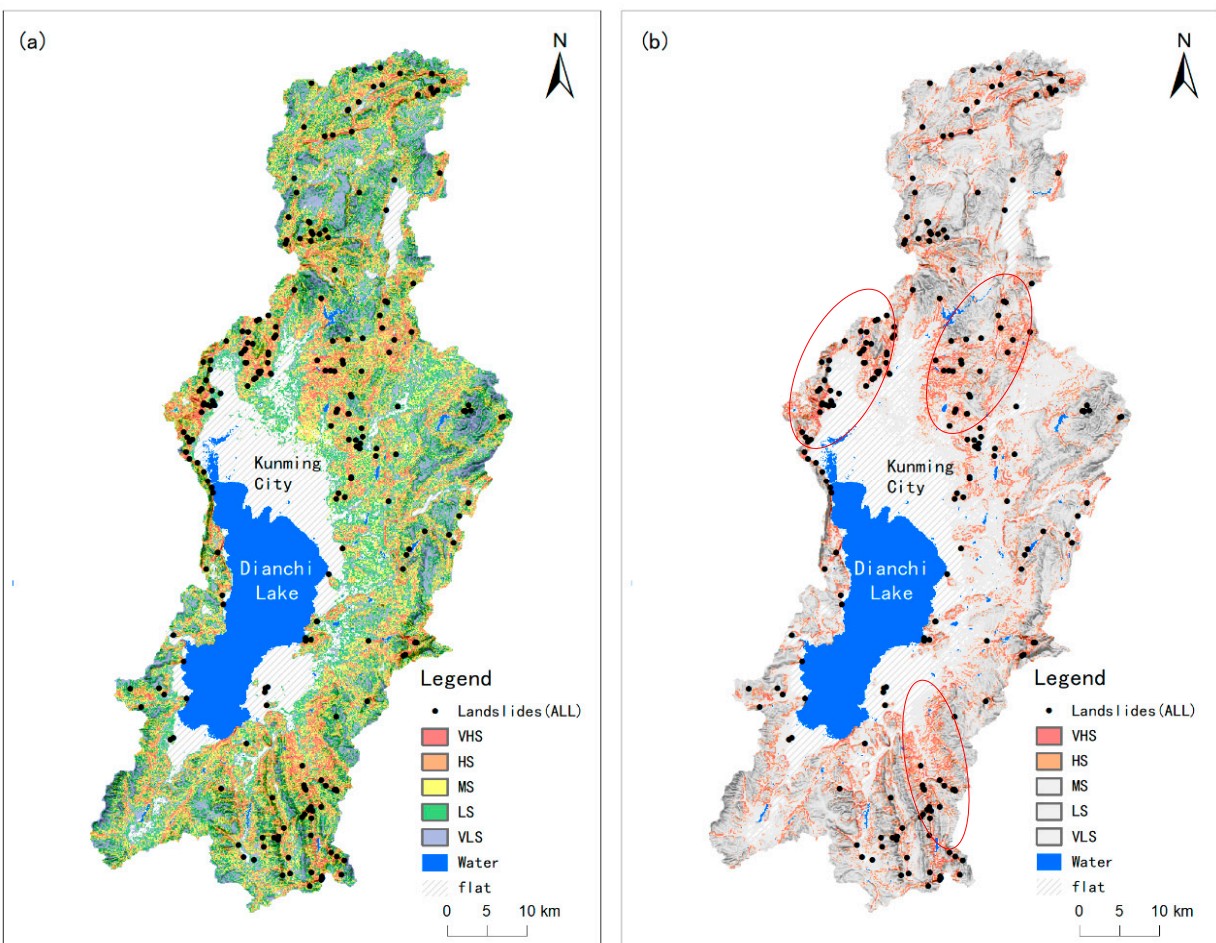

**Figure 27.** Map of susceptibility to landslides based on model M11 and trn. The model M11 has the highest rate of accuracy and validity; (**a**,**b**) are compiled using the same susceptibility partition data. The differences are as follows: (**b**) MS, LS, and VLS use the same general gray color to highlight VHS and HS; the bottom picture is rendered using elevation and hill shade; the red ellipse roughly delineates the areas of high susceptibility and contiguous distribution.

## 5. Discussion

### 5.1. Landslide Susceptibility Zoning and Disaster Prevention Deployment Strategy

Based on the above work, we compiled the landslide susceptibility map of the Dianchi Lake watershed, which has great practical significance. This map provides spatial planners with basic information relating to landslide disasters. It can be used to determine the regional priority for further investigation, support the local planning activities of regional geological disaster prevention and ecological restoration, or create a regional landslide risk exposure assessment. The latter can evaluate the existing elements with landslide risk or those still under planning.

The landslide susceptibility map developed in this paper can effectively predict known and unknown landslides. The fitting accuracy and prediction accuracy of the best model M11 are both ~0.87, and the model coincidence is excellent (Figures 25 and 26). Moreover, ROC_M_trn2TST and the range of ROC_M_trn2trn are closely coincident (Figures 25 and 26), indicating that there is no over-fitting or under-fitting. When only 19.58% of the study area is defined as a high-susceptibility area (VHS + HS), the model can predict 80% of the landslides (Figure 27, Table 2). The above analysis results are satisfactory for the Dianchi Lake watershed.

The map of landslide susceptibility compiled in this paper reveals that the area of high susceptibility (VHS + HS) is relatively large, accounting for about 20% of the study

area (excluding the area with flat and water surface areas), which shows that the natural landslide susceptibility in Dianchi Lake watershed is relatively strong, which poses a great challenge to the comprehensive prevention and control of geological disasters, and this work has a long way to go. In particular, there are large areas of high susceptibility (VHS + HS) in the mountainous area on the edge of the northern basin of the Kunming urban area, and it is almost contiguous. These areas are close to Kunming city and Dianchi Lake, which have a great influence on urban safety and Dianchi Lake water protection and should be regarded as the key areas for landslide prevention and control. Another area with high susceptibility (VHS + HS) is in the southeast of the study area, and mitigation and preventative activities should also be taken in this area.

### 5.2. Important Factors of Landslide Susceptibility and High Sensitivity and Disaster Prevention Suggestions

AUCs (AUC_ALL, AUC_trn, AUC_trn2trn, AUC_trn2TST) of single factors quantify the sensitivity (spatial correlation) of each factor to the impact of landslide, and the evidence weight of single factors (WoE_ALL, WoE_trn) reveals the impact of each classification on the spatial distribution of landslide, while sC defines the significance of the difference between classifications. AUCs, WoEs, and sCs are meaningful indexes by which to quantify the sensitivity of landslide impact.

We have identified more reliable landslide control factors. The results (Figures 9–21) show thirteen factors with AUC $\geq$ 0.6: dRD, HANDV, NDVIlog, SL, RSP, TRI, Rou, Lth, dF, HANDH, Cprof, dCN, and CLCD (Figure 22). The best landslide susceptibility model represents a combination of 11 factors: dRD, HANDV, NDVIlog, SL, RSP, TRI, Rou, Lth, dF, HANDH, and dCN. In the process of step-by-step modeling, according to ROC_M evaluation, Cprof, dCN, and CLCD were rejected because they did not contribute to the explanatory power of the model.

The above results suggest that we should pay attention to the natural conditions and human factors represented by dRD, HANDV, NDVIlog, SL, RSP, TRI, Rou, Lth, dF, HANDH, dCN, CProf, and CLCD, coordinate prevention with planning, construction, and protection, and reduce the induction of landslides.

We also noticed which classification of the above-mentioned important factors is more conducive to the occurrence of landslides. We should pay attention to the slope stability support within 100 m on both sides of the roads and reduce the development in steep slope areas (25–40°), areas where the height difference between the two sides of the stream is 13–67 m, and areas with low vegetation coverage. Attention should also be paid to the preservation and protection of forest vegetation, and the construction planning area should avoid weak rocks such as the affected areas of fault zones (within 121 m on both sides of the faults) and shale siltstone.

### 5.3. The Landslide Susceptibility Evaluation Based on the WoE Method May Be Improved

The optimized classification process sets the classification value based on the nearly continuous cumulative sC curve of evidence weight distribution. This sub-process can capture the trend of evidence weight distribution, overcome the discontinuity of evidence weight distribution in traditional methods, improve the discrimination of landslide sensitivity of each factor, and reduce the subjectivity of factor classification.

The uncertainty analysis obtained via sub-sampling cross-validation technology enables us to verify the weighted uncertainty sampling process related to the introduced error [6]. trn and TST are spatial random sub-samples of the same size from the same dataset, ALL, which represent the same spatial distribution but have different mean sampling errors (MSE) related to sample size [4]. The model performance evaluation based on TST, which is smaller than TRN, must take this into account in order to correctly interpret the model analysis results [31]. MSE based on trn defines the uncertainty of model performance. If the model is well-summarized and there is no obvious over-fitting, then the ROC curve and AUC value should both fall within the MSE range when the model is evaluated

against corresponding TST [4]. Therefore, compared with the traditional no-sampling process (all landslide data are used for analysis), our process is advantageous because the potential impact of random sub-sampling is considered.

We have compared the accuracy and prediction performance of fourteen models with different combinations of factors. The optimal model M11 contains Rou, TRI, and SL with Pearson's C index > 0.7, but the ROC_M_trn2TST of the model not only has no over-fitting, it also shows excellent coincidence. We think that it is not appropriate to exclude the modeling factors only according to Pearson's C index, and it may be more feasible to determine the Cramer's V index and ROC_M comprehensively.

The comprehensive process proposed in this paper combines many techniques, such as optimized classification, cross-validation, and step-by-step modeling, and obtains the model with high accuracy and predictive performance, which shows that this process has good practical value, may improve landslide susceptibility evaluation based on the WoE method, and is worthy of further promotion and application in similar areas.

## 6. Conclusions

(1) The comprehensive process of LSA proposed in this paper has good adaptability, which made a new contribution to the improvement of LSA based on the WoE method. The single-factor categorization optimization sub-process is driven by data, which reduces the subjectivity of factor classification. Cross-validation technology and single-factor WoE statistics reduces the impact of the spatial random effect on factor weight. An effective model was established, and the AUC of fitting and prediction reached 0.8. Cross-validation proves that the model has not been over-fitted.

(2) Eleven factors, namely, dRD, HANDV, NDVIlog, SL, RSP, TRI, Rou, Lth, dF, HANDH, and dCN, were identified as the key factors sensitive to landslides in the study area, which should be considered emphatically in landslide prevention, monitoring, early warning facility layout, and ecological restoration planning.

(3) The area of high susceptibility (VHS + HS) in the Dianchi Lake watershed is large, and the comprehensive prevention of landslides have a long way to go. The large-scale and contiguous high-sensitivity areas in the mountainous areas around the basin have caused serious landslide disasters and degraded the urban safety of Kunming and the water source protection of Dianchi Lake, so it is necessary to strengthen the investigation, monitoring, and risk assessment of landslides.

**Author Contributions:** Conceptualization, G.B., X.Y. and Z.K.; methodology, G.B. and X.Y.; software, G.B. and X.Y.; validation, G.B., X.Y. and Z.K.; formal analysis, G.B. and X.Y.; investigation, G.B., X.Y., Z.K., J.Z., S.Z. and B.S.; resources, G.B., X.Y. and Z.K.; data curation, Z.K. and S.Z.; writing—original draft preparation, G.B., X.Y. and Z.K.; writing—review and editing, G.B., X.Y., Z.K., J.Z., B.S. and S.Z.; visualization, G.B. and X.Y.; supervision, Z.K. and S.Z.; project administration, S.Z. and J.Z.; funding acquisition, S.Z. and J.Z. All authors have read and agreed to the published version of the manuscript.

**Funding:** This research was funded by the science and technology development project of Power China, Sinohydro Foundation Engineering Co., Ltd. (Tianjin, China), the evaluation of rapid excavation of slope cut-off wall in complex geological background area and treatment technology of mud and water inrush in tunnel engineering (Grant No. KKK0202321010); and the scientific and technological development project of Southwest Pipeline Co., Ltd. (Chengdu, China), National Pipe Network Group Research on Hydraulic Protection and Soil and Water Conservation of Oil and Gas Pipelines through Fully Weathered Granite Area (Grant No. KKK0201921153).

**Institutional Review Board Statement:** Not applicable.

**Informed Consent Statement:** Not applicable.

**Data Availability Statement:** The datasets for this study can be obtained by contacting the first author or corresponding author.

**Acknowledgments:** We are very grateful to our colleagues in the team who supported the implementation of this project. We are also sincerely thankful to the editors and reviewers.

**Conflicts of Interest:** The authors declare no conflict of interest.

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
