# Peer review of "Modeling and Assessment of Landslide Susceptibility of Dianchi Lake Watershed in Yunnan Plateau"

_sustainability, doi:10.3390/su152115221_

Round 1
Reviewer 1 Report
Landslides in plateau mountainous areas are always a problem, because they affect people's lives, destroy the land surface and cause economic losses. The findings of this paper provide great guiding significance for deeply understanding the landslide susceptibility of lake watersheds in the study area, and is suitable for publication in the Sustainability.
The paper is well-organized and the facts are separated from interpretations. The manuscript could be reconsidered for publication only if the authors are prepared to improve these shortcomings presented in detail below:
1. Scientific writing: The scientific writing of the manuscript requires revision. I would like to suggest the manuscript to be professionally proofread and edited. Moreover, the authors may pay attention to some aspect of the conventional research writing, especially the connection between the sentences.
2. The summary of the existing research results is illogical, and the scientific problems are not specific. It must contain background, in which problems should be proposed; the present state of the art on the research of the problem, the gap of the present research and the topics to investigate, and objectives of the present research.
3. What is the basis for the selection of the landslide control factors? The author should be introduced in detail in the manuscript.
4. It is suggested to simplify the Results, there are so many data curves that the readers don’t know which is important.
5. The article is more like an engineering report with insufficient theoretical analysis.
6. The innovation point is not prominent enough.
7. A serious Conclusions is needed.
Thanks, and good luck!
Author Response
Thank you very much for taking the time to review our manuscript. Please find the detailed responses in the attachment and the corresponding revisions in the re-submitted files.

Reviewer 2 Report
There is a complete lack of information about Geology, stratigraphy, tectonics, groundwater and lithology
The source of geological data is a map in scale 1:200.000 . The adopted or available scale is not satisfactory for the purpose of the paper.
the same for others adopted thematic layers
In chapter 2.3 do you use a redundant number of parameters (25) for the achievement of landslide dependence. In current literature they do not exceed the number of 10. A large number of potential contributors complicates, as in this paper, evaluation procedure and may introduce some not accuracy in data layers
number 16, slope, has more important significance: stress field
The title repeats 2 times the same words. It can be semplified
Author Response

(The authors gave the same response as above.)

Reviewer 3 Report
1. The introduction lacked the comparison and summary of other scholars' research methods on landslide susceptibility, and it was suggested to supplement.
2. In the introduction(line 53), the Weight of Evidence method was proposed to assess landslide susceptibility, but the reasons and advantages of using WoE were not described in detail.
3. Section 2.1 lacked specific descriptions of regional climate conditions, such as annual precipitation, annual evapotranspiration and annual mean temperature, and it was suggested to supplement.
4. In Section 3.2(line 177), the model improvement method used in the research only relied on adding the factors gradually and compared the obtained index, rather than improving the model itself, which was not innovative to a certain extent.
5. In Section 3.2, why was AUC < 0.59(line 187) chosen as the screening criterion? Suggested to add the cause.
6. In Section 3, the overall design description of the method was too verbose. It was suggested to simplify the language expression and optimize the flow chart.
7. Sections 4.1 and 4.2 only contained simple descriptions of pictures, and lacked a comparative analysis of WoE results for each factor. In addition, section 4.2 included various factors, each factor was a separate picture, and there was no correlation and comparison between different factors.
-
1. In the abstract, the English language tenses were inconsistent, and it was s
uggested unified the tenses. - 2. The language description was not concise enough, such as paragraph 1 in section 5.3, the sentence was too long, and should be improved.
Author Response

(The authors gave the same response as above.)

Round 2
Reviewer 1 Report
The revision was greatly improved according to the reviewer's comments, and it was suggested to be accepted in this version.
Author Response
Thank you very much for taking the time to review our manuscript.
Reviewer 2 Report
The revised form accomplishes the review.
no comment
Author Response

(The authors gave the same response as above.)

Reviewer 3 Report
The drawing in Figure 1 and Figure 2 is not standardized enough, with "N,E" missing in the latitude and longitude range, and the fonts such as Kunming in the picture are too small;
Figure 8 lacks a legend, and it should be noted which quantile each color represents.
In Section 2.2, all landslide points were divided into training and test set, without adding the same number of non-landslide points to constitute samples, resulting in the sample subset containing only landslide features but no non-landslide features. Will the results be affected? Please give the reasons in the original text.
The overall English language needs to be improved, pay attention to brevity.
Author Response
Thank you very much for taking the time to review our manuscript.
Please see the attachment.
